

# STRAPS v1.0: Evaluating a methodology for predicting electron impact ionisation mass spectra for the aerosol mass spectrometer

David .O. Topping[1,2], James. Allan[1,2], M. Rami, Alfarra[1,2], and Bernard Aumont[3]

[1]School of Earth and Environmental Science, University of Manchester, Manchester, M13 9PL, United Kingdom
[2]National Centre for Atmospheric Science, University of Manchester, Manchester, M13 9PL, United Kingdom
[3]LISA, UMR CNRS 7583, Universite Paris Est Creteil et Universite Paris Diderot, Creteil, France

*Correspondence to*: David. O. Topping (david.topping@manchester.ac.uk)

**Abstract.** Our ability to model the chemical and thermodynamic processes that lead to secondary organic aerosol (SOA) formation is thought to be hampered by the complexity of the system. While there are fundamental models now available that can simulate the tens of thousands of reactions thought to take place, validation against experiments is highly challenging. Techniques capable of identifying individual molecules such as chromatography are generally only capable of quantifying a subset of the material present, making it unsuitable for a carbon budget analysis. Integrative analytical methods such as the Aerosol Mass Spectrometer (AMS) are capable of quantifying all mass, but because of their inability to isolate individual molecules, comparisons have been limited to simple data products such as total organic mass and O:C ratio. More detailed comparisons could be made if more of the mass spectral information could be used, but because a discrete inversion of AMS data is not possible, this activity requires a system of predicting mass spectra based on molecular composition.

In this proof of concept study, the ability to train supervised methods to predict electron impact ionisation (EI) mass spectra for the AMS is evaluated. Supervised Training Regression for the Arbitrary Prediction of Spectra (STRAPS), is not built from first principles. A methodology is constructed whereby the presence of specific mass-to-charge ratio (m/z) channels are fit as a function of molecular structure before the relative peak height for each channel is similarly fit using a range of regression methods. The widely-used AMS mass spectral database is used as a basis for this, using unit mass resolution spectra of laboratory standards.

Key to the fitting process is choice of structural information, or molecular fingerprint. Our approach relies on using supervised methods to automatically optimise the relationship between spectral characteristics and these molecular fingerprints. Therefore, any internal mechanisms or instrument features impacting on fragmentation are implicitly accounted for in the fitted model. Whilst one might expect a collection of keys specifically designed according to EI fragmentation principles to offer a robust basis, the suitability of a range of commonly available fingerprints is evaluated.





Initial results suggest the generic public 'MACCS' fingerprints provide the most accurate trained model when combined with both decision trees and random forests with median cosine angles of 0.94-.0.97 between modelled and measured spectra. There is some sensitivity to choice of fingerprint, but most sensitivity is in choice of regression technique. Support Vector Machines perform the worst, with median values of 0.78-0.85 and lower ranges approaching 0.4 depending on the

fingerprint used. More detailed analysis of modelled versus mass spectra demonstrates important composition dependent sensitivities on a compound-by-compound basis. This is further demonstrated when we apply the trained methods to a model α-pinene SOA system, using output from the GECKO-A model. This shows that use of a generic fingerprint referred to as 'FP4' and one designed for vapour pressure predictions ('Nanoolal') give plausible mass spectra, whilst the use of the MACCS keys perform poorly in this application, demonstrating the need for evaluating model performance against other

SOA systems rather than existing laboratory databases on single compounds.

Given the limited number of compounds used within the AMS training dataset, it is difficult to prescribe which combination of approach would lead to a robust generic model across all expected compositions. Nonetheless, the study demonstrates the use of a methodology that would be improved with more training data and data from simple mixed systems for further

validation. To facilitate further development of the method, including application to other instruments, the model code for re-training is provided via a public Github and Zenodo software repository.

# 1 Introduction

Volatile organic compounds (VOCs), emitted from both natural and anthropogenic sources, are oxidised in the atmosphere to form lower-volatility species that condense onto aerosol particles or contribute to new particle formation (Laaksonen et al.,

2008;Sipila et al., 2016;Ehn et al., 2014). With an enormous number of species that are present, this diversity in chemistry is reflected in the extensive range of species and chemical signatures identified in ambient studies (Hamilton et al., 2013). Within atmospheric science, it is desirable to develop models for secondary organic aerosol (SOA) formation based on a given set of precursors and photochemical processing. Within most global and regional models, often-used techniques include modelling representative photochemical yields from specific precursors and tuning accordingly (Spracklen et al.,

2011) or employing a parametric model such as the volatility basis set (Robinson et al., 2007). While both of these approaches can deliver realistic absolute concentrations, because they are not based on explicit physical processes, their predictive skill is always subject to question (Hallquist et al., 2009;Bergstrom et al., 2012). It is therefore desirable to develop SOA models based around actual molecular processes and kinetics constrained through laboratory experiments (where available), such that this skill can be evaluated. Such models rely on explicit chemical mechanisms such as the

Master Chemical Mechanism (MCM) (Saunders et al., 1997) (Saunders et al., 1997) or the GECKO model (Aumont et al., 2005). While this mechanistic approach has resulted in poor performance in terms of absolute mass concentrations in the past (Volkamer et al., 2006), much of this shortfall can be accounted for by not considering all precursors (in particular the



semi-volatile and intermediate-volatility organic matter), unexpected processes likely to produce lower-volatility products (e.g. oligomerisation and autoxidation (Ehn et al., 2014) and inadequacies associated with phase partitioning models (Barley and McFiggans, 2010;Valorso et al., 2011;McVay et al., 2016). As the availability of data regarding these has improved and thus our understanding of these processes matured, the performance of the models has become more realistic (McVay et al.,

2016). The development of more applicable explicit models has been facilitated by the ability to automatically predict processes rather than prescribe them (Aumont et al., 2012;Aumont et al., 2005) as has been implemented in the Generator of Explicit Chemistry and Kinetics of Organics in the Atmosphere (GECKO-A) and the forthcoming version 4 of the MCM (http://gotw.nerc.ac.uk/list_full.asp?pcode=NE%2FM013448%2F1). This can be supplemented by the automated prediction of properties important for partitioning, using generalised informatics tools such as UManSysProp (Topping et al., 2016).

While it is unlikely that such complex models would be used directly for large-scale Eularian chemical transport and climate models, and uncertainties with regards to fundamental properties remain (Bilde et al., 2015), they are still highly useful for benchmarking and providing the parameters for simpler models.

Comparison of model output with measurements in the ambient air and in laboratory is required to test model accuracy. With

current analytical methods, it is impossible to detect and quantify every compound in the particle even if we can predict compound-by-compound speciation. While there are techniques capable of resolving a large number of molecules such as electrospray ionisation and two-dimensional gas chromatography (Noziere et al., 2015), comprehensively calibrating for and thus providing quantitative data on the abundances of the molecules is difficult. The AMS, which is often used in chamber and flow tube experiments, is capable of delivering data on the total mass concentration of organic matter and some other

simple top-down metrics such as the O:C ratio (Aiken et al., 2007) however this does not provide the ideal constraint of such models.

While the mass spectral data can be further investigated through inspection of markers at specific m/z channels (such as 43 and 44) (Ng et al., 2011), such data tends to be qualitative and result in speculative conclusions (Morgan et al., 2010). In

theory, the data across the mass spectrum could be more systematically compared with the modelled data if knowledge of the instrument response to molecular features could be invoked in a general fashion. (Ehn et al., 2014)

In this proof of concept study we evaluate a methodology to bridge existing model measurement comparison. A database of the AMS mass spectral responses to various molecules has been built up over the years and this has been used to characterise

the response of certain key peaks to certain functional groups (Ulbrich et al., 2009;Ehn et al., 2014). In this study we use that information to develop and evaluate regression software that predicts an AMS spectrum based on the predicted aerosol composition (figure 1)



This is not the first study on predicting EI mass spectra based on molecular composition, or to demonstrate the potential for predicting instrument response functions(Camredon et al., 2007). Bauer and Grimmer (2016) recently reviewed the current performance of quantum chemistry methodologies in predicting EI mass spectrometry for small to medium sized molecules from first principles. Whilst that study documents improving general applicability, they are not immediately suitable for

5   predicting AMS mass spectra because the thermal desorption promotes further fragmentation and, in some cases, pyrolysis(Canagaratna et al., 2015). While the standard AMS analysis takes these processes into account through empirical calibrations, the exact physical processes taking place within the vaporiser system are still the subject of considerable debate (Murphy, 2016;Drewnick et al., 2015;Robinson et al., 2016), so the bottom-up modelling of this is not possible with the current state of knowledge.

Distinct from all previous approaches, the approach presented here relies on supervised learning methods to automatically optimise the relationship between spectral characteristics and molecular features from the instrument in question. Therefore, any internal mechanisms or instrument features impacting on fragmentation are implicitly accounted for in the fitted model.

In section 2 the methodology behind constructing a predictive model is presented, whereas section 3 focuses on results regarding the accuracy of a model with respect to comparisons with spectra for individual components. In addition we present results from simulating the mass spectra of α-pinene aerosol using the GECKO-A model before we discuss future data requirements in section 4.

## 2 Methodology

Figure 1 displays the workflow used in building the predictive model. First, a model is trained to predict the occurrence of specific m/z channels as a function of molecular composition before a model for each m/z channel is trained to predict peak height within that channel. It is worthwhile detailing the molecular information used to train each model. Each molecule has varying levels of structural features, the combination of which provides each molecule with a 'fingerprint'. Numerically, this fingerprint can be thought of as a collection of stoichiometric information for a collection of distinct features. For

example, for a collection of 10 compounds we would construct a matrix of stoichiometric information where each row represents a specific molecule and each column the stoichiometry of a given feature. We new refer to each column as a 'key', which might be a specific functional group or feature associated with that molecule. We retain the use of the word 'key' since it can provide more generic information that a functional group. To re-iterate, the entire row we refer to as the molecular fingerprint. For example, identifying the occurrence of carboxylic acid groups is a key within the AIOMFAC

fingerprint (Zuend et al., 2011). We then take this information and use it to train a model to predict both the occurrence of a specific m/z channel and then peak heights.



The underlying physical principles of EI (F. W. McLafferty, 1994) adjusted to the AMS (Gasteiger et al., 1992), do not exist in algorithmic form, so there is currently no a priori basis for choosing the most appropriate fingerprint for this work. Therefore, a collection of common fingerprints are tested in this study and their performance critically evaluated. This is an important sensitivity since one might expect a collection of keys that relate to EI fragmentation principles to offer a robust

basis for fitting any method used here. We discuss this further in section 5.

Fingerprints used in this study include those employed in activity coefficient and vapour pressure predictive techniques provided by the UManSysProp package (Topping et al., 2016;Zuend et al., 2011;Nannoolal et al., 2008), alongside more general fingerprints including the MACCS keys and FP4 keys (Putta et al., 2003). It is difficult to find information on

provenance behind these latter generic fingerprints (Putta et al., 2003), other that they are designed to cover a set of molecular features that would be used across a broad range of applications. The MACCS fingerprint provides up-to 162 unique keys of any given molecule, the FP4 fingerprint featuring up to 320. The current implementation of the MACCS keys from the Pybel package (O'Boyle et al., 2011) is used whereas the FP4 keys are extracted from the RDKit open source informatics package (http://www.rdkit.org/docs/index.html). Each key is represented in the UManSysProp package (Topping

et al., 2016) using SMARTS notation, and each molecule using the SMILES format. The matrix of keys used to fit each method is constructed by systematically parsing each molecule. Figure 2 demonstrates the use of the MACCS SMARTS to populate a matrix of keys. The full collection of SMARTS keys can be found in the source code. Please refer to section 5 on code availability.

With regards to the supervised methods used, an ensemble tree is trained to predict the occurrence of specific m/z channels as a function of any given fingerprint. To predict peak height per m/z channel, we evaluate a number of supervised methods available in the SciKit-learn package: Generalised Linear methods, Support Vector Machines [with 3 separate kernels], Stochastic Gradient Descent, Bayesian Ridge, Ordinary Least Squares, Decision Trees and Ensemble methods (Pedregosa et al., 2011). There are a number of other methods available, yet as we will discuss in section 5, the results from this study

demonstrate a potential whilst further data is needed to confirm general applicability, including the use of other methods. For a brief overview of each method, we refer the reader to Ruske (2016) and references therein. Before training each method, the matrix of identified keys were standardized between zero and one using the MinMaxScaler pre-processing feature within the Scikit learn package. In addition, the use of variable selection is designed to use only those features deemed important to construct fingerprint-peak height relationships to try and mitigate any under or over fitting. The sensitivity to these

procedures are discussed in section 3.2 To compare modelled and measured mass spectra, the cosine angle from a dot product of the two are used, focusing on specific m/z channels that are typically found as features within atmospheric and smog chamber mass spectra (Ulbrich et al., 2009): 15,18,28,29,39,41,43,44,50,51,53,55,57,60,73,77,91.



The ability of each method to replicate the entire database is first evaluated. Whilst training on a subset and comparing with the entire database will test wider applicability, this initial comparison quantifies the appropriateness of the different fingerprints in building an accurate model.

## 3. Results

### 3.1 Sensitivity to choice of molecular fingerprint

Figure 3 visually compares the number of keys extracted from the 100 compounds in the AMS library according to choice of fingerprint. Data is presented according to the use of AIOMFAC [bottom left], MACCS [top left], Nanoolal [bottom right] and FP4 [top right] keys. Using the AIOMFAC fingerprint leads to, at most, 17 keys identified from the AMS library. The Nanoolal fingerprint leads to a larger set of keys (19), with the MACCS fingerprint providing the most (74) and the FP4 keys

the second highest (30). The use of more or less information in the fitting procedure should not be assumed to automatically lead to a more accurate predictive model. Ideally there should be a balance between the number of features identified and how those features relate to the mechanisms of fragmentation on the molecule within the instrument in question. As we have already noted, comparing the information provided by each fingerprint with a working knowledge of the mechanics of EI fragmentation might help understand why a given fingerprint is more suitable. However we first and foremost wish to

demonstrate the feasibility of using pre-defined fingerprints with prescribing any dependencies, and the exact physical processes taking place within instrument are still the subject of considerable debate.

Table 1 presents the median cosine angle of modelled spectra fit to the entire AMS database derived from the different supervised methods and different fingerprints, to 2 decimal places. The left hand sided box-plots in figure 4a-d display the

20 entire cosine angle spread for each method for both the MACCS (4a), FP4 (4b), AIOMFAC (4c) and Nanoolal fingerprints (4d). When fitting to the entire library of AMS spectra, initial results suggest that the tree-based methods ['Tree','Forest'] perform better than others, with the MACCS keys leading to improved model performance over other fingerprints. However, the difference between using either the MACCS or Nanoolal keys, for example, is not significant for any given supervised method as noted in Table 1. Rather than demonstrating 100% accuracy, the values of 1.00 must be taken with

25 caution as we demonstrate in proceeding analyses. Whichever fingerprint is used, the ranking of performance between supervised methods remains similar, with the tree-based methods, Ordinary Least Squares and Bayesian Ridge outperforming Stochastic Gradient Descent and all Support Vector Machine kernels. Along with higher median values, the spread of cosine angles from the tree based methods and Ordinary Least Squares is much lower than all other methods. Whilst the use of MACCS and FP4 provide, in theory, more information, there is some similarity in structural information

provided in all keys. For example, each fingerprint identifies key functional groups such as alkanes, alcohol, ketones etc, whilst the FP4 and MACCS keys in particular include more positional detail including relative positions of groups. At least for the 100 compounds in the AMS library, that additional information leads to a slight increase in cosine angle agreement of



around 0.02 between methods, if we use only results from table 1 and figure 4. We discuss the significance of values displayed in table 1 after performance is re-evaluated following a more general approach of training to a subset of compounds, and the use of variable selection, in the next section.

5 **3.2 Training to a subset and variable selection**

Table 2 presents the median cosine angle between modelled and predicted mass spectra, as a function of fingerprint and regression technique, when training to a subset of the entire database and use of variable selection. To minimise over fitting any model to specific features, the process of variable selection allows us to refit the model to those keys deemed most important. The combination of both strategies might be considered the most suitable test of the methodology presented, with
10 the full spread of statistics presented in the right hand column of figures 4a-d,. It should be noted that randomly selecting the subset used for training leads to a significant decrease in model performance. This is due to missing keys within the training subset that are deemed important in predicting spectra for those compounds outside of the subset. A different approach is to select the subset according to maximising the number of keys across each molecule in the training subset, and is used in our proceeding analysis.

In some cases, such as with the OLS and Forest methods, the data provided in Table 2 suggests that using both strategies leads to a lower median cosine angle, thus slightly reduced model performance. However, in practice, the statistics presented in Table 1 should not be considered a true test of the methodology, but rather a precursor demonstration of the sensitivity to choice of fingerprint, and perhaps any variability in instrument response across the AMS library.

On the significance of the value of cosine angle, Figures 5 and 6 display predicted spectra for compounds not included in the training set, along with the cosine angle between modelled and measured spectra. For Oxalic acid in Figure 5, the difference in performance between the FP4 and MACCS fingerprint [cosine of 0.83 and 0.77] is apparent through certain features, including the relative proportion of peak heights for the 3 dominate channels, and the ratio of f44 to f43. In Figure 6, a
25 similar pattern is found for Leucine, including a marked difference in whether the model predicted non-zero entries across f41 – f44. Whilst a small subset, these results suggest use of the cosine angle alone is not sufficient to validate model performance, which is confirmed in section 3.3 when applied to the α-pinene system. Based on these comparisons, a tentative suggestion of using a cosine angle of 0.8 might go some way to clarifying the performance statistics provided in Tables 1 and 2 and Figure 4. Indeed, results demonstrate that, whilst statistics in Table 2 and Figure 4 suggest similar
30 performance for both MACCS and FP4 keys, this performance is composition dependent. This reflects sensitivity to information used in the training process and how similarity between performances should be taken with caution in prescribing which method to take forward. This is better highlighted in the proceeding section with regards to a model SOA





system. Results at least suggest the tree based methods are at least the most stable given the higher range of cosine angles presented in Figure 4 and the decision tree method will be used in all proceeding analysis.

**3.3 Example application to a model aerosol system.**

In this section we apply the trained methods to a model SOA system, using output from the GECKO-A model used by

5 (Valorso et al., 2011) to study SOA formation from α-pinene in a simulated chamber experiment. The purpose of this exercise is to explore sensitivity of predicted mass spectra to combined speciated output from a fixed model configuration through varying fingerprints to support the comparisons made in the previous section. It is not designed as a thorough quantitative analysis of spectra comparisons, but rather to demonstrate the ability to extract specific features and highlight sensitivities to choice of model configuration. A recent study of McVay et al. (2016) presented results demonstrating

sensitivity to processes included in that model, including the addition of autoxidation mechanisms, for example. They proposed that autoxidation might resolve some or all of measurement–model discrepancy, but that this hypothesis could not be confirmed until more explicit mechanisms are established for α-pinene autoxidation(McVay et al., 2016). One might imagine an ideal sensitivity study would be to use speciated output from these updated models and add additional constraint to prescribing model performance through a comparison between measured and predicted mass spectra. Indeed, that is a

rationale behind the study presented here. However, as proceeding results will demonstrate, with the existing training data and lack of validation on simple mixtures, there is potential for false positives in the predicted spectra to confuse a diagnosis of accurate model configurations. Specifically, the composition space derived from a series of box-model configurations would need to be mapped onto the existing space covered by the AMS spectral library. Combined with additional measurements of mixed systems of known composition, we could then prescribe a more robust set of regression model

configurations through which a more detailed sensitivity study could take place.

Nonetheless, to illustrate sensitivity to choice of fingerprint in a complex system, Figure 7 displays the predicted mass spectra for the GECKO-A model results of Valorso et al. (2011) combined with the experimental data taken from a chamber-based α-pinene SOA formation experiment reported by Alfarra et al. (2013) (high VOC:NO$_x$ ratio). Without further

refinement of model and measurement conditions, these results exhibit large errors in the predicted mass spectra when using MACCS keys, despite the brief analysis presented in section 3.2. This demonstrates that over fitting to distinct features in the training set and difference between this composition space and that provided by the box-model output are leading to features that are missed in the final spectra. This is further supported by the abundance of features extracted from the training set displayed in figure 3.

To expand on this performance, Figure 8 displays the predicted mass spectra f44 peak height versus O:C ratio from the GECKO-A model results of Valorso et al (2011) in a manner similar to Aiken et al. (2008). There are 9 points on each curve, representing points in time during the GECKO-A simulation, with the model predicting a monotonic increase in O:C over





time. It is worth noting the value values are low compared to typical atmospheric LV-OOA (Aiken et al., 2008;Kroll et al., 2011). Overall, use of the FP4 and Nanoolal keys give absolute f44s that compare well with published calibrations relative to O:C, specifically Aiken et al. (2008) and the updated calibration presented by Canagaratna et al. (2015). The direction of the trend in f44 versus O:C is reversed when using the Nanoolal keys, with f44 decreasing with O:C, which runs contrary to

expectations. However, it should be noted that the values are within the spread of values used to generate the Aiken et al. (2008) and Canagaratna et al. (2015) calibrations, as these performed regressions over much bigger ranges of O:C than obtained in this simulation, so the prediction based on Nanoolal keys could still be plausible.

Figure 9 displays the predicted f44 to f43 peak heights from the model system using the commonly used 'triangle plot'

(Morgan et al., 2010;Ng et al., 2011), compared with the experimental data taken from the chamber experiments of Alfarra et al. (2013) and also Chhabra et al. (2011), who studied the formation of α-pinene oxidation in response to different oxidants. Note the trajectories in this space are not monotonic for either the experimental or simulated data, which indicate the complexities in interpreting spectra based on these metrics. Results suggest that f43 values when using the FP4 and Nanoolal keys are plausible when compared to published studies. The f44 peak height is systematically low for all fingerprints, as also

shown in figure 5-7. However, rather than a deficiency in the mass spectral prediction methods, this is likely due to a deficiency in the Valorso et al. (2011) model treatment. It has recently been shown how important mechanisms such as autooxidation are to the α-pinene SOA system (Ehn et al., 2014), which are capable of rapidly adding oxygenated functional groups to the molecules that are responsible for both the suppression of vapour pressures necessary for SOA formation and also the increase in the f44 metric (Canagaratna et al., 2015). More recent versions of GECKO-A have included such

mechanisms (McVay et al., 2016), however a systematic comparison of the predicted spectra based on these inclusions is beyond the scope of this proof-of-concept paper and will be presented in a future publication.

### 4. Discussion and future work

The preceding analysis demonstrates the potential for the methodology presented to lead to interesting investigations on

model versus measured mass spectra. However, there are a number of remaining improvements that need to be made. It is inevitable that not all of the chemical species predicted by the models will be covered by previous laboratory work. If a class of species predicted by any chemical mechanism is identified as not covered by existing SMARTS-based fragmentation rules, it could be characterised in the laboratory using the same facilities and methodologies employed for previous characterisation work (Canagaratna et al., (2015) and references therein).

The methods here have a number of uses, although it must be re-iterated that the predicted mass spectra are not definitive. The performance of this method will be improved by the addition of further training data. As outlined in the introduction, the ability of this model to predict AMS spectra will be useful in the development and validation of explicit SOA mechanisms in





the laboratory, meaning that the models can be challenged by the entire mass spectrum and not just the mass and O:C ratio. This method can also be used at the experiment design stage, allowing predictions of whether an AMS will be able to discern expected changes in composition associated with a process and thus whether it will be useful to test particular hypotheses.

The method could also be used to simulate atmospheric aerosol, probably if the chemical model is used in a Lagrangian configuration. In addition to the insights gained in atmospheric processes, this could be used to critically test the data model used in positive matrix factorisation (PMF) (Ulbrich et al., 2009). Because of the condition that PMF factors have fixed profiles, the reduction of the complexity associated with atmospheric SOA to (typically) two factors results in an increase in 'rotational ambiguity' associated with the factorisation. A two-component factorisation of SOA is often interpreted as

representing the 'low volatility' and 'semivolalite' components of the SOA (Jimenez et al., 2009), although this has shown not to be applicable to all environments, where other sources of variability contribute to the split in the factors (Young et al., 2015). If the mass spectral response to atmospheric SOA could be more explicitly simulated using this technique, a synthetic AMS dataset could be used as the subject of PMF analysis in a manner similar to Ulbrich et al. (2009). This in turn could be used to investigate the contributions of the factorisation on a more explicit level and investigate the effects this has on

rotational ambiguity and the validity of solutions.

**5. Code availability**

A publicly available copy of the code used to derive performance statistics of the chosen regression methods can be found at : https://github.com/loftytopping/STRAPS covered by a GPL v3.0 license. This includes a copy of the AMS spectral files

that now also include appropriate SMILEs strings. We also provide an associated DOI for the exact model version given in this paper as provided by the Zenodo service: https://zenodo.org/record/213068#.WFlryyiPD3s

Please note that an extension to the SMARTS libraries included in UmanSysProp was carried out in this project. To review the features extracted for each fingerprint, please refer to the files 'FP4.smarts', 'MACCS.smarts', 'nannoolal_primary.smarts' and 'aiomfac_unifac.smarts' included in the directory

*UManSysProp_public/umansysprop/data/.*

**Author contributions:**

David Topping concieved the methdology presented and performed the subsequent model development and analysis. James Allan and Rami Alfarra offered expert guided constraints on evaluating the model results, including selecting the best

comparison metrics to use. Bernard Aumont supplied the results from the Valorso et al (2011) study. All authors contributed to the writing of the manuscript.

**Acknowledgements**: David Topping, James Allan and Rami Alfarra received funding from the National Centre for Atmospheric Science [NCAS]. This work was built on informatics developed under NERC grant NE/H002588/1.





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



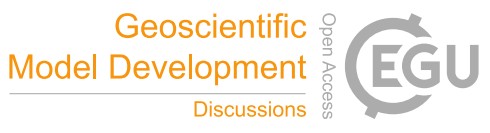

| Method | MACCS | FP4 | AIOMFAC | Nanoolal |
|---|---|---|---|---|
| SVM RBF | 0.87 | 0.85 | 0.86 | 0.85 |
| SVM Poly | 0.84 | 0.83 | 0.82 | 0.81 |
| SVM Lin | 0.80 | 0.80 | 0.79 | 0.79 |
| BRR | 0.94 | 0.92 | 0.90 | 0.91 |
| OLS | 1.00 | 0.96 | 0.94 | 0.94 |
| SGDR | 0.88 | 0.82 | 0.80 | 0.80 |
| Tree | 1.00 | 1.00 | 1.00 | 1.00 |
| Forest | 1.00 | 1.00 | 1.00 | 1.00 |

Table1 - Median cosine angle between measured and predicted spectra when fitting to the entire dataset as a function of molecular fingerprint [Given above each column]. The method labels are as follows: SMV [Support vector Machine with 3 kernels (RBF, Poly[nomial] and Lin[near]), BRR: Bayesian Ridge, OLS: Ordinary Least Squares, SGDR:Stochastic Gradient Descent, Tree: Decision Tree and Forest: Random Forest.

| Method | MACCS | FP4 | AIOMFAC | Nanoolal |
|---|---|---|---|---|
| SVM RBF | 0.85 | 0.82 | 0.80 | 0.81 |
| SVM Poly | 0.82 | 0.81 | 0.81 | 0.79 |
| SVM Lin | 0.78 | 0.79 | 0.78 | 0.78 |
| BRR | 0.93 | 0.91 | 0.88 | 0.88 |
| OLS | 0.95 | 0.93 | 0.90 | 0.90 |
| SGDR | 0.87 | 0.82 | 0.81 | 0.80 |
| Tree | 0.97 | 0.97 | 0.94 | 0.96 |
| Forest | 0.97 | 0.97 | 0.95 | 0.96 |

Table 2 - Median cosine angle between measured and predicted spectra, when using 80% of the compounds in the training process, with variable selection, as a function of molecular fingerprint [Given above each column]. The method labels are as follows: SMV [Support vector Machine with 3 kernels (RBF, Poly[nomial] and Lin[near]), BRR: Bayesian Ridge, OLS: Ordinary Least Squares, SGDR:Stochastic Gradient Descent, Tree: Decision Tree and Forest: Random Forest.

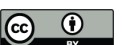



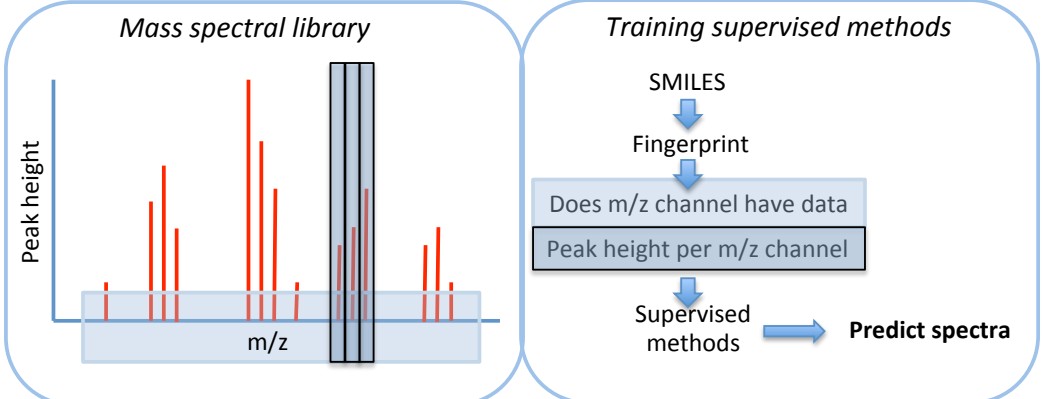

**Figure 1 – Schematic of workflow used in the training process. For a normalised mass spectrum, the SMILEs string associated with each compound is combined with a given molecular fingerprint to train methods to predict the occurrence of a given m/z channel and then a peak height.**

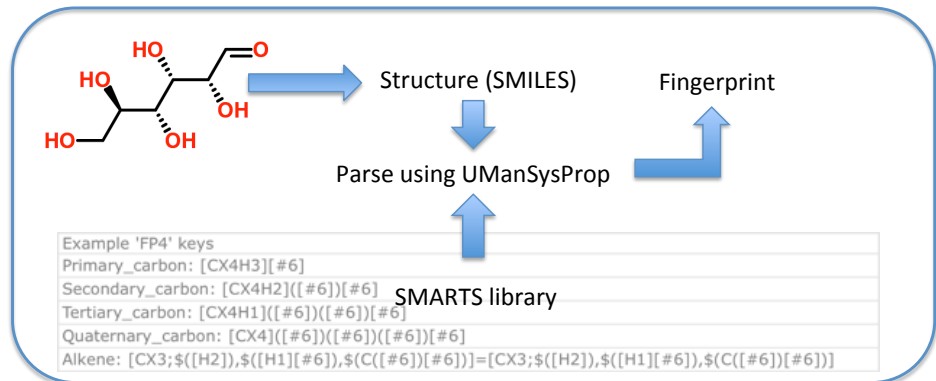

**Figure 2. Basic schematic of interrogating a SMILES string with a SMARTS library to construct a molecular fingerprint.**




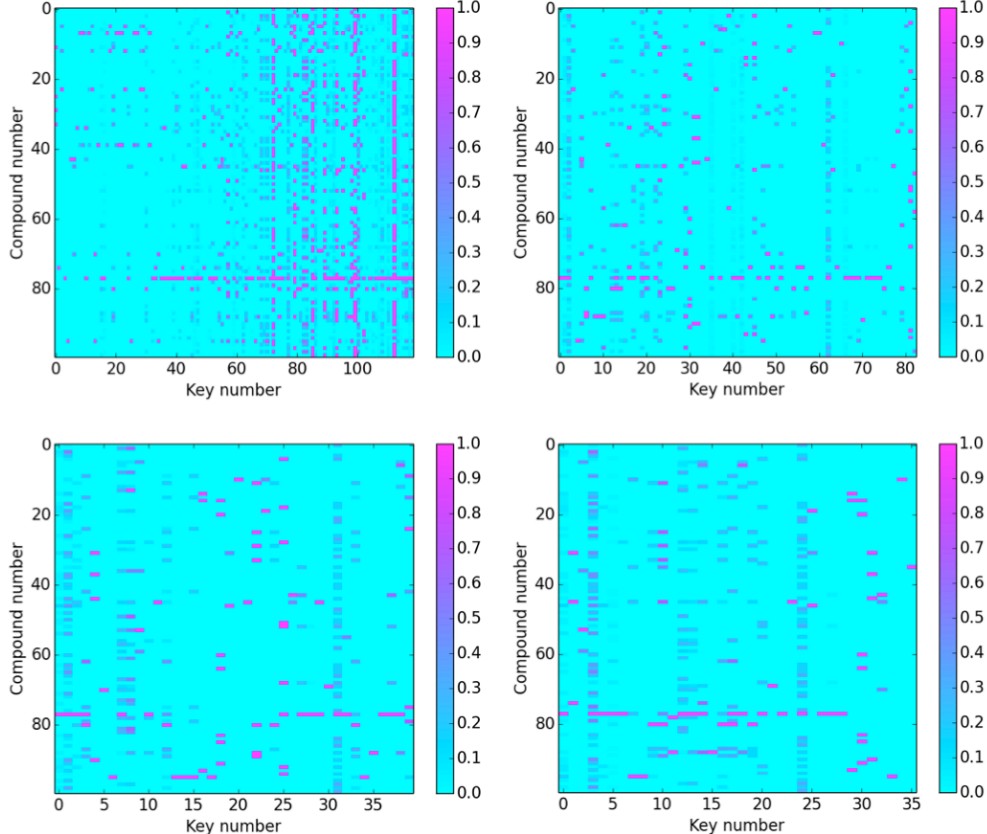

**Figure 3 – Sparsity of keys extracted (x axes) from each compound (y axes) as a function of molecular fingerprint used (Top left: MACCS, Top right: FP4, Bottom left: AIOMFAC, Bottom right: Nanoolal). Keys are coloured according to normalised stoichiometry across all compounds.**



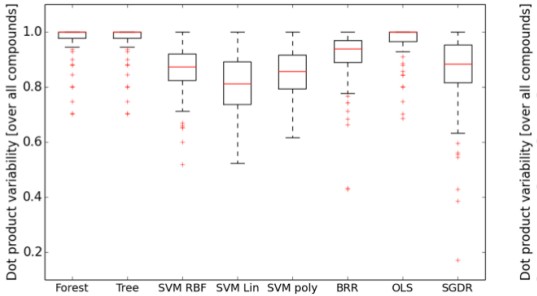 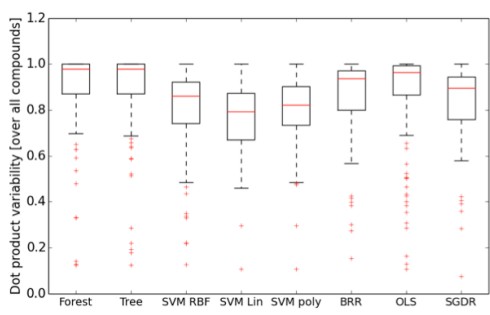

**Figure 4a – Spread of cosine angle between experimental and predicted mass spectra [y axes] for all 100 compounds in the AMS library as a function of supervised method [x axes] using the MACCS fingerprint. left: using all compounds in the training process. right: using 80% of the compounds in the training process with variable selection. The method labels are as follows: SMV [Support vector Machine with 3 kernels (RBF, Poly[nomial] and Lin[near]), BRR: Bayesian Ridge, OLS: Ordinary Least Squares, SGDR:Stochastic Gradient Descent, Tree: Decision Tree and Forest: Random Forest.**

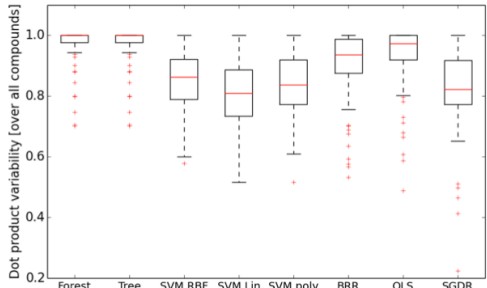 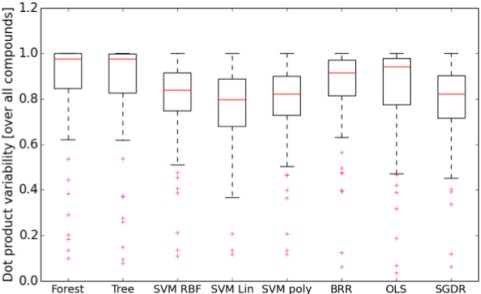

**Figure 4b – Spread of cosine angle between experimental and predicted mass spectra [y axes] for all 100 compounds in the AMS library as a function of supervised method [x axes] using the FP4 fingerprint. left: using all compounds in the training process. right: using 80% of the compounds in the training process with variable selection. The method labels are as follows: SMV [Support vector Machine with 3 kernels (RBF, Poly[nomial] and Lin[near]), BRR: Bayesian Ridge, OLS: Ordinary Least Squares, SGDR:Stochastic Gradient Descent, Tree: Decision Tree and Forest: Random Forest.**





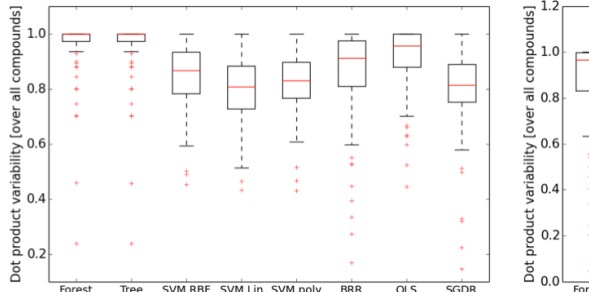
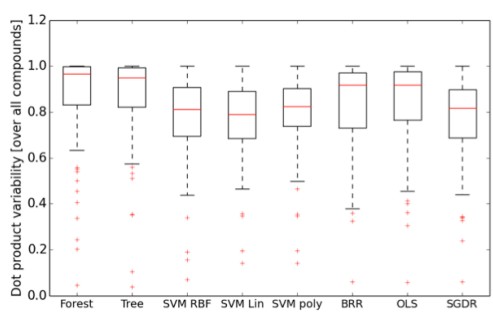

**Figure 4c – Spread of cosine angle between experimental and predicted mass spectra [y axes] for all 100 compounds in the AMS library as a function of supervised method [x axes] using the AIOMFAC fingerprint. left: using all compounds in the training process. right: using 80% of the compounds in the training process with variable selection.**
5  **The method labels are as follows: SMV [Support vector Machine with 3 kernels (RBF, Poly[nomial] and Lin[near]), BRR: Bayesian Ridge, OLS: Ordinary Least Squares, SGDR:Stochastic Gradient Descent, Tree: Decision Tree and Forest: Random Forest.**

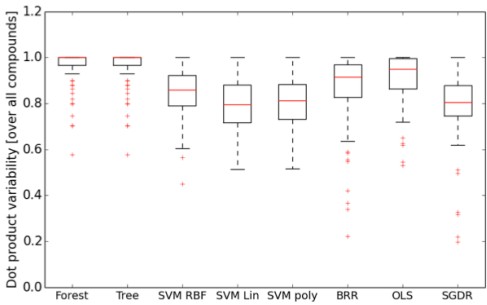
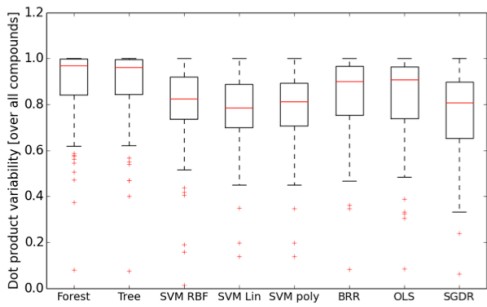

**Figure 4d – Spread of cosine angle between experimental and predicted mass spectra [y axes] for all 100 compounds**
10  **in the AMS library as a function of supervised method [x axes] using the Nanoolal fingerprint. left: using all compounds in the training process. right: using 80% of the compounds in the training process with variable selection. The method labels are as follows: SMV [Support vector Machine with 3 kernels (RBF, Poly[nomial] and Lin[near]), BRR: Bayesian Ridge, OLS: Ordinary Least Squares, SGDR:Stochastic Gradient Descent, Tree: Decision Tree and Forest: Random Forest.**





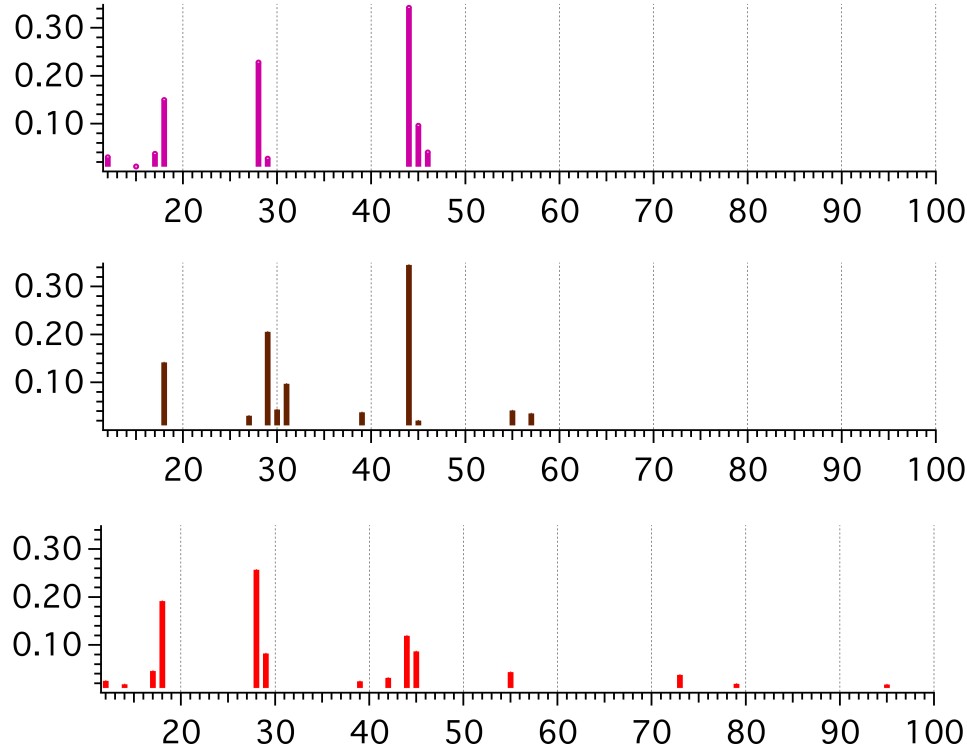

**Figure 5 – Measured mass spectra for Oxalic acid [top] versus predicted mass spectra from an ensemble tree using the FP4 fingerprint [middle, cosine of 0.83] and the MACCS fingerprint [bottom, cosine of 0.77].**





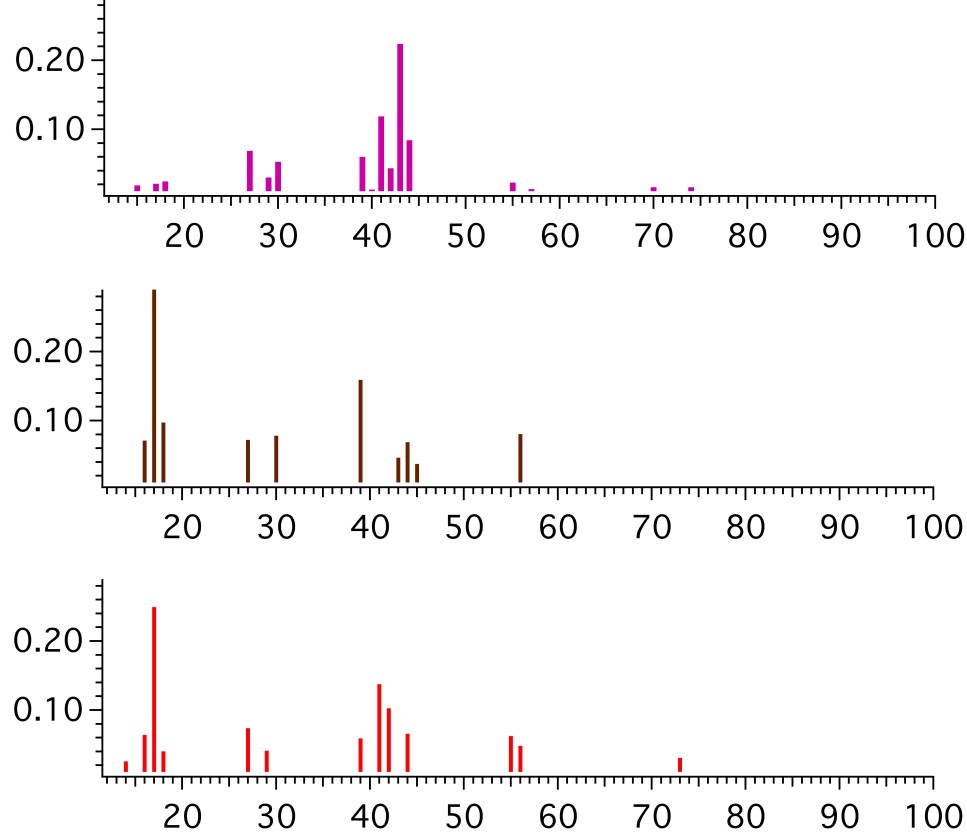

**Figure 6 – Measured mass spectra for Leucine [top] versus predicted mass spectra from an ensemble tree using the FP4 fingerprint [middle, cosine of 0.70] and the MACCS fingerprint [bottom, cosine of 0.94].**





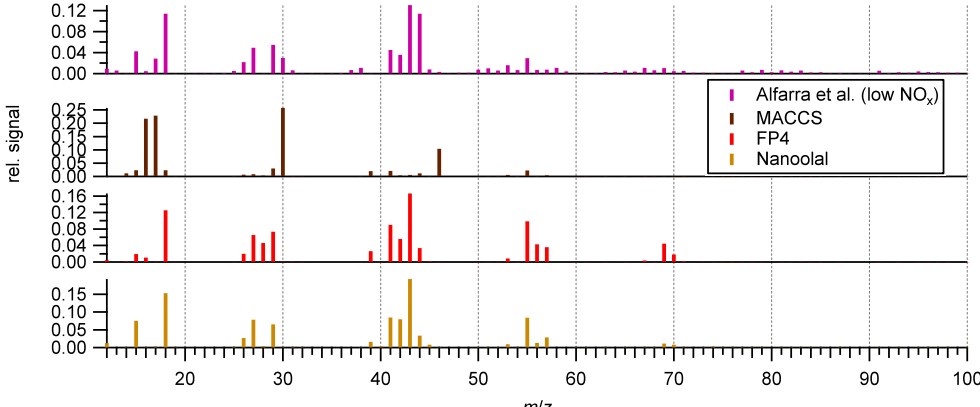

**Figure 7 – Comparison of the predicted mass spectra of α-pinene SOA based on the GECKO-A simulation presented by Valorso et al. (2011) using various fingerprinting techniques. These are compared with an actual α-pinene SOA mass spectrum obtained by Alfarra et al. (2013) during a chamber experiment.**





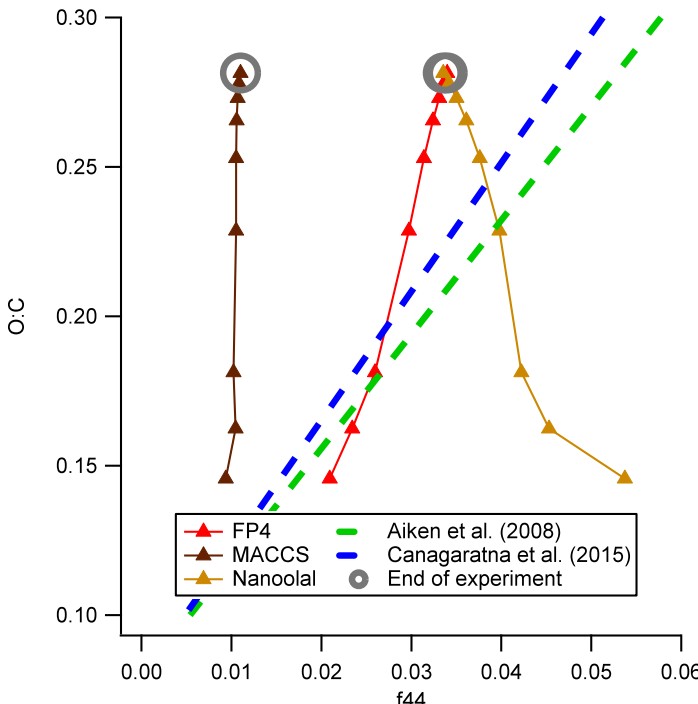

**Figure 8 Comparison of O:C ratios and predicted fractional contribution to the AMS m/z 44 channel (f44) for the Valorso et al. (2011) GECKO-A simulation, compared against the regressions performed by Aiken et al. (2008) and Canagaratna et al. (2015). The highlighted points indicate the final points in the simulation.**





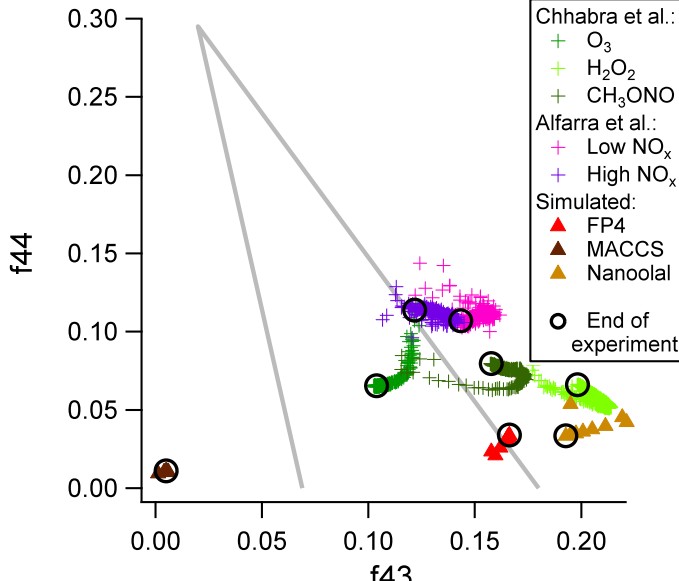

**Figure 9 – 'Triangle plot' comparing predicted f44 and f43 values for the Valorso et al. (2011) GECKO-A α-pinene SOA simulation with chamber experiments. The Chhabra et al. (2011) data compares different oxidant systems and is taken from figure 2A of that paper. The chronological final points in each dataset are highlighted.**

