# Peer review of "STRAPS v1.0: Evaluating a methodology for predicting electron impact ionisation mass spectra for the aerosol mass spectrometer"

_Geoscientific Model Development, 2016_

## Referee Comment (RC1) · Anonymous Referee #1 · 17 Feb 2017

In this work, the authors harvest a set of molecular descriptors from various molecules and establish relationships to mass fragments observed by aerosol mass spectrometry. They evaluate different sets of molecular descriptors and supervised learning methods to evaluate the range of predictive capability that can be achieved for a set of individual compounds, and also for mixtures derived from chemically explicit simulation. While inherent differences between model simulations and reality preclude strict comparisons, they also evaluate the general trajectory in the evolution of the predicted f44 to simulated O:C ratio and f44 to f43. The idea presented in this manuscript is a nice one. A link between chemical composition and AMS mass spectra would be desirable; the challenges for predicting electron impact mass spectra from first principles and justifications for taking a chemoinformatic/statistical approach are outlined well. Statistical models relating molecular properties to kinetic rate constants (structure-activity relationships) are widely accepted in the community (e.g., Carl et al., 2007), so an effort such as this one relating molecular properties to observable instrumental signals are a welcome addition. The manuscript is recommended for publication by Geoscientific Model Development; addressing the following comments may improve the readability of the manuscript.

The authors highlight the "proof-of-concept" nature of the study with many issues to be resolved in future work, which is understandable given its novelty. However, the main achievements in this work are not highlighted well. Is the MACCS fingerprints most successful just because of the sheer number of keys, each of which contribute to predictions, or are there particular structural elements not present in the others that improve the predictions? The generally poor performance of SVMs for all keys is surprising, is it possibly due to the high dimensionality in the underlying representations that is not present in the others, or is there a more obvious reason to the authors?

How are the tuning parameters for the model parameters determined? For instance, the penalty factor for SVM, etc.?

Are cosine angles (uncentered correlations) sufficient to capture agreement that represents more than the range (minimum and maximum) relative ion counts for each spectrum? This angle may not represent disagreement in relative ion counts that are of intermediate value very well. In that there is precedent for cosine angles for mass spectra comparison, it is a safe metric, but the authors may look at analyzing residuals for each mass fragment to understand what their model gets right and less right (to generalize on illustrations provided in Figures 5 and 6, which are incidentally missing axes labels). There is some mention about f43 being somewhat reasonable and f44 being underpredicted, but this seems a bit buried in the presentation.

Is the number of keys used vs. m/z variables and issue? Given the smaller number

of non-zero keys and number of samples, is it reasonable to try to predict 300 m/z's in the AMS spectrum (In Figures 5-8 only 100 are shown, but is the model trained only to predict 100 m/z's)? Would not the authors benefit from trying to reproduce a "reduced" set of spectra (e.g., reconstructed from a truncated set of PCA or PMF components)? Is there a reason why all keys were not combined into a single fingerprint? It would be simple to remove redundant keys simply by inspection, if that were a concern.

Regarding the comparison of f44 and O:C (Figure 8), is not the COO+ associated with m/z 44 more sensitive to dicarboxylic acids (Russell et al., 2009)?

A minor point: The simulation (photoxidation) conditions of Valorso (2011) can be repeated in the caption of Figure 9 so the reader can immediately contextualize the comparison.

References:

S. A. Carl, L. Vereecken, and J. Peeters, Phys. Chem. Chem. Phys., 2007, 9, 4071-4084 doi: 10.1039/B705505F.

L. M. Russell, R. Bahadur, L. N. Hawkins, J. Allan, D. Baumgardner, P. K. Quinn, T. S. Bates, Organic aerosol characterization by complementary measurements of chemical bonds and molecular fragments, 2009, 43, 6100–6105, doi:10.1016/j.atmosenv.2009.09.036.

———————————————

---

## Referee Comment (RC2) · Anonymous Referee #2 · 18 Feb 2017

General Comments

It is acknowledged that techniques are required to help illuminate the highly complex chemical and physical properties and processes associated with SOA formation, evolution and composition. One such considerably powerful instrumental technique is Aerosol Mass Spectrometry (AMS), now used in many areas of atmospheric, climate and air quality science. As stated by the authors, AMS is able to capture all mass but unfortunately is unable to provide the required high level of speciation; thus generally a cumbersome ensemble of techniques is required to fully test measurements with state-of-the-art models. A method to help improve enhanced chemical speciation of AMS spectral composition, and hence empower model vs. measurement comparisons

to enhance fundamental understanding, would be welcomed.

In this work the authors present a proof of concept study around the use of trained supervisionary regression methods to predict the composition of AMS spectra of organic aerosol components. The authors present a well thought out series of sequential results comparing model outputs using various supervised learning methods with experimental data for single component spectra and more complex chemically driven spectra. The authors provide a direct link to the code developed.

This work is very novel and has importance for model and mechanism development in the laboratory and in ambient air (e.g. model vs. measurement comparison for elucidation of fundamental processes); it also has the potential to add to the widely used Positive Matrix Factorisation (PMF) technique and to be employed by other analytical methods. As such it is recommended that this manuscript be published in Geoscientific Model Development subject to the authors addressing the following Specific and Technical Comments.

Specific Comments

P2, Line 2: Could the authors state the rationale for choosing to assess performance simply with cosine angles?

P2, Line 15: could the authors suggest which other analytical techniques could potentially benefit from the method and include appropriate references?

P4, Section 2: It is unclear from this text whether the fingerprint is in fact a single mass spectrum of m/z vs abundance. Can this be section be rephrased slightly with a more explicit statement, please? Further to this, should each column refer to a given m/z, is this enough information, or are other concomitant spectral features required to validate the presence of a certain functional group? If each single column "key" is able to contain sufficient information, can the authors clarify and appropriately state this in the text (further to references e.g. Ulbrich et al.)?

P6, Section 3.1: Can the authors comment on the sensitivity of the technique to the various functional groups listed on line 30, for example? Are there any inherent instrumental sensitivity issues with certain functional groups that might limit the effectiveness of the technique at a top level?

Owing to composition dependence acknowledged by the authors, it would be nice to see additional data c.f. Figures 5 and 6, for other single precursors. Are these data available?

P7, Lines 30 – 33: Regarding the statement – "This reflects sensitivity to information used in the training process and how similarity between performances should be taken with caution in prescribing which method to take forward", as this represents a limitation, could the authors expand their discussion slightly, i.e. potential magnitude of uncertainty associated with inaccurate method prescription? Further, could the authors clarify the sensitivity of the technique to user required experience and expertise?

P8, Lines 18 – 20: When the authors refer to addition of data from mixed systems, are they referring to an ensemble photo-oxidation study, or simply an inert multicomponent mixture? Did the authors consider a test intermediate in complexity, e.g. the obvious intermediate between a single compound mass spectrum and a chamber photo-oxidation experiment would be an analysis of a mixture of 2-3 compounds, without the complex oxidative chemistry. Was this considered?

Regarding the AMS data employed (e.g. Figure 7): How were these data treated? Were they experiment averaged, summed, normalised? Despite the reference to Alfarra et al., 2013, it may be useful to briefly state this on introduction of the experimental data in order to provide context.

Please check reference formatting throughout, e.g. spaces between text and parentheses and improper use of chronological ordering of multiple citations.

Technical Corrections

[Figure]

P2, Line 21: Please add more indicative primary source references; this paper is rather specific

P2, Line 30: Reference repeated

P3, Line 14: "...air and in THE laboratory..."

P4, Line 26: "now" rather than "new"

P4, Line 28: "than" rather than 'that"?

P5, Line 10: "than" rather than "that"

P5, Line 30: Full-stop missing after "3.2"

P6, Lines 14 – 16: Rewrite to facilitate ease of reading

P8, Lines 9 – 11: Sentence is awkward, I suggest it is rewritten for clarity

P8, Line 21: "Fingerprints"

P9, Line 1: Repeated word - "value values"

P14, Tables 1 and 2 legends: Right [ parenthesis missing

P17, Figures 4: axis labels are too small and potentially unreadable in final print, please increase the text size

P17, Figures 4 legends: Right [ parenthesis missing

P19, Figure 5: Axis labels missing

P20, Figure 6: Axis labels missing

---

## Author Comment (AC1) · 11 Apr 2017

We would like to thank the reviewer for their recognition of the potential of the approach presented here. In the following we respond to all comments, including detailing some additional work that has been carried out with regards to fingerprint analysis. In the following response we separate and number all distinct comments in order of their appearance in the review, highlighting new text added to the manuscript where appropriate.

1) *Is the MACCS fingerprints most successful just because of the sheer number of keys, each of which contribute to predictions, or are there particular structural elements not present in the others that improve the predictions?*

**Response:** Comparing average performance statistics in section 3.1 at first implies this might be the case. However the comparison with spectra from the Alfarra et al. (2013) paper illustrates the MACCS keys perform poorly. Interrogating the performance from predictions using the MACCS keys for specific compounds illustrates a few problems that might reflect a lack of generality across the MACCS keys. For example, the FP4 keys cycle through systematic functional groupings such as: primary carbon, secondary carbon, tertiary carbon...primary alcohol, secondary alcohol, tertiary alcohol etc. This would lead to a maximum of 320 keys per molecule. MACCS keys on the other hand are almost seemingly designed to capture a random, although extensive, set of features leading to a maximum of 162 features for any given molecule. As we note in the manuscript, it is difficult to find the provenance behind the MACCS keys. However, we have added the following text in section 2, page 5, to try and clarify the issue [new text presented in italics]: '*There are some common features between each fingerprint library, but also a range of differences. For example, all libraries identify the presence of the CH2 group, but then differ in optional connecting groups. The FP4 keys cycle through systematic groupings, such as: primary carbon, secondary carbon, tertiary carbon...primary alcohol, secondary alcohol, tertiary alcohol etc. Similar groups are detected using the activity coefficient and vapour pressure keys.* The full collection of SMARTS keys can be found in the source code and we discuss suggestions for future work on refining fingerprints in section 4. Please refer to section 5 on code availability.'

2). *The generally poor performance of SVMs for all keys is surprising, is it possibly due to the high dimensionality in the underlying representations that is not present in the others, or is there a more obvious reason to the authors?*

**Response:** We agree this is surprising, especially given the extent of applications to which SVMs are applied. At first we assumed this was down to how the data was normalized prior to training. However, using a maximum/minimum scalar prior to training did not improve performance. There are differences according to which kernel is used. It might be true that dimension reduction procedures, such as PCA, might improve performance. With this in mind, we have conducted tests on using PCA prior to training, using the combined set of fingerprints as requested in point '6' addressed shortly. Based on these results we have added an additional table [table 3] demonstrating the effect of dimension reduction procedures on the performance of all methods, using the combined fingerprint approach:

| Method | 20 | 10 | 8 | 4 |
|---|---|---|---|---|
| SVM RBF | 0.84 | 0.84 | 0.85 | 0.67 |
| SVM Poly | 0.83 | 0.83 | 0.81 | 0.79 |
| SVM Lin | 0.80 | 0.80 | 0.80 | 0.80 |
| BRR | 0.93 | 0.90 | 0.89 | 0.87 |
| OLS | 0.94 | 0.89 | 0.89 | 0.87 |
| SGDR | 0.89 | 0.89 | 0.89 | 0.88 |
| Tree | 0.98 | 0.98 | 0.98 | 0.98 |
| Forest | 0.99 | 0.99 | 0.99 | 0.99 |

*Table 3 - Median cosine angle between measured and predicted spectra, applying PCA analysis to the 'combined' fingerprints, as a function of the number of principal components used given above each column. The method labels are as follows: SMV [Support vector Machine with 3 kernels (RBF, Poly[nomial] and Lin[near])], BRR: Bayesian Ridge, OLS: Ordinary Least Squares, SGDR:Stochastic Gradient Descent, Tree: Decision Tree and Forest: Random Forest.*

We have also added the following text to section 3.2 [new text presented in italics], which is renamed to: 3.2 Training to a subset, variable selection **and dimension reductions**. 'in practice, the statistics presented in Table 1 should not be considered a true test of the methodology, but rather a precursor demonstration of the sensitivity to choice of fingerprint, and perhaps any variability in instrument response across the AMS library. *On this, the use of the 'combined' fingerprint demonstrates the ability to retain information from those keys that improve overall performance. Given their wide use across many disciplines, it is difficult to quantify the reasons behind the poor performance of the Support Vector Machines relative to other methods. To assess whether dimension reduction procedures would improve accuracy, table 3 presents the median and overall spread of cosine angles when using Principal Component Analysis (PCA) on the 'combined' fingerprints. The number of principal components was varied between 20, 10, 8 and 4. Generally, reducing the number of keys from, up to, 278 to 20 components, leads to an improvement of around 0.01-0.02 in all methods apart from Ordinary Least Squares and Support Vector Machines with both the polynomial and linear kernels. Results demonstrate clear sensitivity to the number of components when combined with the RBF Support Vector Machine kernel, performance varying from 0.84 to 0.67 on reducing the number of components from 20 to 4.*'

We cannot say with any certainty what the true cause of variability within each regression technique is. Ultimately, we feel this proof of concept study needs building on with appropriate laboratory data before further quantification of dependencies would be possible. Whilst we state the rationale in the original manuscript, we have added the following text in section 4 to re-iterate this: '*On the sensitivity to choice of fingerprint, our results demonstrate compound specific trends that lead to performance variability when applied to a complex SOA system that is not apparent when analysing median cosine angle statistics. Combining available fingerprints into one can slightly improve performance in some cases, but as the comparison of isolated MACCS versus FP4 performance illustrates, there is potential danger in over fitting to distinct features in the training set that is not provided by the box-model output. To re-iterate, one might expect a collection of keys that relate to EI fragmentation principles to offer a more robust basis for fitting any method used here. However, that requires further work with additional laboratory data to validate the efficacy of any new bespoke fingerprint.*'

*3) How are the tuning parameters for the model parameters determined? For instance, the penalty factor for SVM, etc.?*

**Response:** Using the cosine angle between spectra as a measure of good fit, parameters for each method, where required, are cycled until the most effective combination were found. These parameter ranges are presented in the code release and are specific to each algorithm,.

4) *Are cosine angles (uncentered correlations) sufficient to capture agreement that represents more than the range (minimum and maximum) relative ion counts for each spectrum? This angle may not represent disagreement in relative ion counts that are of intermediate value very well. In that there is precedent for cosine angles for mass spectra comparison, it is a safe metric, but the authors may look at analyzing residuals for each mass fragment to understand what their model gets right and less right (to generalize on illustrations provided in Figures 5 and 6, which are incidentally missing axes labels). There is some mention about f43 being somewhat reasonable and f44 being under predicted, but this seems a bit buried in the presentation.*

**Response:** There are indeed other metrics we could have employed to measure distance between mass spectra, however we considered cosine to be the most appropriate. Firstly, because our aim is to replicate the AMS instrument response function, which can be modelled as a linear addition of multiple component mass spectra, we reason that it would make the most sense to use a metric that places linear weight on the peaks' relative intensities. Secondly, while a different metric may place a relatively greater weight on intermediate peaks (thus ensuring a more general agreement over a larger number of peaks), we would have to take care not to also unduly weight the minor peaks, which can be problematic. As such, an element of subjectivity would have been introduced in the choice of algorithm, which in itself would require more testing. It is possible that there is a better closeness metric that could be tested as part of future work and this would be easily testable within the STRAPS framework, however see that as outside the scope of this particular paper. Concerning the comparison between f43 and f44, this refers to the specific comparison between the GECKO-A run and roughly comparable chamber experiments, however we must stress that this test was only to demonstrate proof-of-concept and not perform a systematic comparison to assess the performance. We merely show that the values produced for these two common AMS metrics are plausible in magnitude. For this to be done properly, a chemical model run matched to the exact chamber system should be performed with a state-of-the-art model; this will form part of future work and a full, systematic comparison of peak magnitudes will be performed there.

5) *is it reasonable to try to predict 300 m/z's in the AMS spectrum (In Figures 5-8 only 100 are shown, but is the model trained only to predict 100 m/z's)? Would not the authors benefit from trying to reproduce a "reduced" set of spectra (e.g., reconstructed from a truncated set of PCA or PMF components)?*

**Response:** The methodology presented here is based on predicting a response for each channel, and then predicting the peak height for each channel. Each m/z therefore has its own model and there is not dependency on whether 100, 150 or 300 m/z's are chosen. There is no penalty to predicting the high m/z peaks, as these generally represent a low mass fraction and contribute little to the cosine of the comparisons. However, there will be a tangible disadvantage to operating on a reduced dataset because the data reduction in itself will inherently remove information that is possibly of value for training, so there is a very real risk of an inferior training.

6) *Is there a reason why all keys were not combined into a single fingerprint? It would be simple to remove redundant keys simply by inspection, if that were a concern. Regarding the comparison of f44 and O:C (Figure 8), is not the COO+ associated with m/z 44 more sensitive to dicarboxylic acids (Russell et al., 2009)?*

**Response:** This is a good point, and we have conducted additional simulations to investigate this. It is worth noting the initial aim of the paper was to illustrate the use of 'standard' fingerprint libraries, as they exist as distinct developments. As noted in the manuscript, ideally we would like to take this proof of concept work forward by constructing a library of keys that better represents the mechanism of fragmentation within the AMS. It might be that converting general rules of EI fragmentation would be a useful starting point. Tables 1-2 now includes median cosine angles from each regression technique when combining all keys into one fingerprint:

| Method | MACCS | FP4 | AIOMFAC | Nanoolal | Combined |
|---|---|---|---|---|---|
| SVM RBF | 0.87 | 0.85 | 0.86 | 0.85 | 0.85 |
| SVM Poly | 0.84 | 0.83 | 0.82 | 0.81 | 0.83 |
| SVM Lin | 0.80 | 0.80 | 0.79 | 0.79 | 0.80 |
| BRR | 0.94 | 0.92 | 0.90 | 0.91 | 0.95 |
| OLS | 1.00 | 0.96 | 0.94 | 0.94 | 0.99 |
| SGDR | 0.88 | 0.82 | 0.80 | 0.80 | 0.89 |
| Tree | 1.00 | 1.00 | 1.00 | 1.00 | 1.00 |
| Forest | 1.00 | 1.00 | 1.00 | 1.00 | 1.00 |

*Table1 - Median cosine angle between measured and predicted spectra when fitting to the entire dataset as a function of molecular fingerprint [Given above each column]. Please note, the term 'Combined' refers to a combination of all individual fingerprints into one. The method labels are as follows: SMV [Support vector Machine with 3 kernels (RBF, Poly[nomial] and Lin[near])], BRR: Bayesian Ridge, OLS: Ordinary Least Squares, SGDR:Stochastic Gradient Descent, Tree: Decision Tree and Forest: Random Forest.*

| Method | MACCS | FP4 | AIOMFAC | Nanoolal | Combined |
|---|---|---|---|---|---|
| SVM RBF | 0.85 | 0.82 | 0.80 | 0.81 | 0.85 |
| SVM Poly | 0.82 | 0.81 | 0.81 | 0.79 | 0.82 |
| SVM Lin | 0.78 | 0.79 | 0.78 | 0.78 | 0.80 |
| BRR | 0.93 | 0.91 | 0.88 | 0.88 | 0.94 |
| OLS | 0.95 | 0.93 | 0.90 | 0.90 | 0.98 |
| SGDR | 0.87 | 0.82 | 0.81 | 0.80 | 0.88 |
| Tree | 0.97 | 0.97 | 0.94 | 0.96 | 0.98 |
| Forest | 0.97 | 0.97 | 0.95 | 0.96 | 0.98 |

*Table 2 - Median cosine angle between measured and predicted spectra, using 80% of the compounds in the training process, with variable selection, as a function of molecular fingerprint [Given above each column]. Please note, the term 'Combined' refers to a combination of all individual fingerprints into one. The method labels are as follows: SMV [Support vector Machine with 3 kernels (RBF, Poly[nomial] and Lin[near])], BRR: Bayesian Ridge, OLS: Ordinary Least Squares, SGDR:Stochastic Gradient Descent, Tree: Decision Tree and Forest: Random Forest.*

We have also added the following text to section 3.1, Page6 [new text in italic]: 'Table 1 presents the median cosine angle of modelled spectra fit to the entire AMS database derived from the different supervised methods and different fingerprints, *either isolated or combined into one*, to 2 decimal places.'

Followed by: '*A key objective of this study, noted above, is to demonstrate the use of pre-defined fingerprints in constructing a predictive model. However, it is useful to also demonstrate the efficacy of combining the information from each fingerprint into one, without relating variable performance according to physical processes taking place within the instrument. The performance of combining all fingerprints into one, represented in table 1 under the column heading 'combined', illustrates a similar trend in performance between methods.*'

This is now combined with the request presented earlier to assess the role of dimension reductions, using PCA, leading to a new table [3] and subsequent text presented in response to point 2. We also add the following text to the final paragraph in the abstract [new text in italic]:' the study demonstrates the use of a methodology that would be improved with more training data, *fingerprints designed explicitly for fragmentation mechanisms occurring within the AMS,* and data from additional mixed systems for further validation.'

Whilst these new simulations add an interesting angle, we still need more experimental data to resolve any issues with over or under fitting that might occur using our limited, and yet, somewhat disparate set of compounds in the present training database. We feel this is one reason the MACCS keys perform so poorly when methods are applied to the outputs of Valorso et al (2011), in that there are specific keys that are leading to over fitting to the training dataset.

7) *A minor point: The simulation (photoxidation) conditions of Valorso (2011) can be repeated in the caption of Figure 9 so the reader can immediately contextualize the comparison.*

**Response:** This has now been added to the figure caption. Concerning the comparison between f43 and f44, this refers to the specific comparison between the GECKO-A run and roughly comparable chamber experiments, however we must stress that this test was only to demonstrate proof-of-concept and not perform a systematic comparison to assess the performance. We merely show that the values produced for these two common AMS metrics are plausible in magnitude. For this to be done properly, a chemical model run matched to the exact chamber system should be performed with a state-of-the-art model; this will form part of future work and a full, systematic comparison of peak magnitudes will be performed there.

---

## Author Comment (AC2) · 11 Apr 2017

We would like to thank the reviewer for their recognition of the novelty of the approach presented here. In the following we separate and number all distinct comments in order of their appearance in the review, highlighting new text added to the manuscript where appropriate.

Specific Comments
*1) P2, Line 2: Could the authors state the rationale for choosing to assess performance simply with cosine angles?*

**Response:** We are happy to clarify this, as in response to the other referee. There are indeed other metrics we could have employed to measure distance between mass spectra, however we considered cosine to be the most appropriate. Firstly, because our aim is to replicate the AMS instrument response function, which can be modelled as a linear addition of multiple component mass spectra, we reason that it would make the most sense to use a metric that places linear weight on the peaks' relative intensities. Secondly, while a different metric may place a relatively greater weight on intermediate peaks (thus ensuring a more general agreement over a larger number of peaks), we would have to take care not to also unduly weight the minor peaks, which can be problematic. As such, an element of subjectivity would have been introduced in the choice of algorithm, which in itself would require more testing. It is possible that there is a better closeness metric that could be tested as part of future work and this would be easily testable within the STRAPS framework, however see that as outside the scope of this particular paper.

*2) P2, Line 15: could the authors suggest which other analytical techniques could potentially benefit from the method and include appropriate references?*

**Response:** It is possible that other techniques may benefit from this, but this is specifically tested around an instrument that gives ensemble data in response to a liner addition of signatures. Other forms of mass spectrometry, such laser desorption and ionisation and electrospray ionisation, suffer from matrix effects, so the model will need further development for this. It will also be of limited use for 'soft' ionisation techniques where there is little molecular fragmentation (such as chemical ionisation mass spectrometry), as the components will mainly be intact molecular ions (or adducts) that will require no training. However, it could be useful in interrogating poorly-resolved mixtures in gas chromatography mass spectrometry. It may be very powerful when applied to spectroscopic techniques such and nuclear magnetic resonance spectroscopy or Fourier transform infrared spectroscopy.

*3) P4, Section 2: It is unclear from this text whether the fingerprint is in fact a single mass spectrum of m/z vs abundance. Can this be section be rephrased slightly with a more explicit statement, please?*

**Response:** Apologies. We have replaced the sentence **'Each molecule has varying levels of structural features, the combination of which provides each molecule with a 'fingerprint''** with **'***Each molecule has varying levels of structural features, which can be written in terms of a 'fingerprint'. This fingerprint is a numerical identification of a given structure that can equally be thought of as stoichiometric information...***'**

*4) Further to this, should each column refer to a given m/z, is this enough information, or are other concomitant spectral features required to validate the presence of a certain functional group? If each single column "key" is able to contain sufficient information, can the authors clarify and appropriately state this in the text (further to references e.g. Ulbrich et al.)?*

**Response:** We apologize for any lack of clarity here. The collection of molecules, represented as SMILES strings, is parsed to produce a matrix where each column represents the stoichiometry of a particular key, or feature. This entire matrix is used to fit a predict model for each m/z channel.

We have added the following, similar, text to the end of page 4 to attempt clarification of this procedure: *'To re-iterate, in constructing a model that can predict AMS mass spectra, a library of compounds with measured spectra are used to train a series of regression techniques. This collection of molecules, represented as SMILES strings, is parsed to produce a matrix where each column represents the stoichiometry of a particular key, or feature. This entire matrix is used to fit a predict model for each m/z channel.'*

*5) P6, Section 3.1: Can the authors comment on the sensitivity of the technique to the various functional groups listed on line 30, for example? Are there any inherent instrumental sensitivity issues with certain functional groups that might limit the effectiveness of the technique at a top level?*

**Response:** The fact that AMS and even EI in general has issues with certain functional groups (see cited literature, in particular Canagaratna et al., 2015) is well documented. Examples include the overlap of multiple functional groups at m/z=43 and the tendency for multifunctional molecules to generate a large signal at m/z=44. However, providing 'top-down' rules for this would be inherently difficult and it is for this exact reason that we chose to test the technique using pre-existing fingerprinting techniques and objectively determine their comparative performance. With further work, it may be possible to develop an AMS-specific fingerprinting technique based on instrument knowledge and compare this against the conventional fingerprinting techniques, however one must take care not to base the fingerprinting technique too closely on the laboratory data that will subsequently be used for training, as this will introduce an element of confirmation bias and thus may give false confidence in the fitting and subsequent extrapolations.

*6) Owing to composition dependence acknowledged by the authors, it would be nice to see additional data c.f. Figures 5 and 6, for other single precursors. Are these data available?*

**Response:** We agree this would be very useful. Firstly, we feel that recommendations for additional data described in section 4 should be pursued before a detailed analysis of additional precursors systems. Secondly, the state of box-models used to study multiple precursors is highly variable and not particularly well documented or with a common data/software repository. The recent study of McVay et al (2015) might improve predictions presented in figure 7-9 due to additional mechanisms such as the formation of HOMs. However, the presentation of other box-model suggests the requirement for tracking each compound in the condensed phase, to be used as input into STRAPS, is not necessarily followed. Given the two commonly used chemical mechanisms, the Master Chemical Mechanism (MCM) and GECKO, carry individual molecular representations as SMILES strings, this would not take much work to improve. It would be a very useful development to have a central repository of box-model output that is visible and easy to access.

*7) P7, Lines 30 – 33: Regarding the statement – "This reflects sensitivity to information used in the training process and how similarity between performances should be taken with caution in prescribing which method to take forward", as this represents a limitation, could the authors expand their discussion slightly, i.e. potential magnitude of uncertainty associated with*

*inaccurate method prescription? Further, could the authors clarify the sensitivity of the technique to user required experience and expertise?*

**Response:** Regarding the first point, we cannot at this stage prescribe a magnitude of uncertainty for any given method without further testing. To re-iterate the recommended data requirements presented and extended in section 4, it would be highly useful to obtain additional laboratory data on systems from a specific series of compounds to enable this quantification. Regarding the second point, we would hope that the use of openly available libraries in the Scikit learn package, and fully documented software repositories, will enable anyone to replicate or extend the work presented here.

*8) P8, Lines 18 – 20: When the authors refer to addition of data from mixed systems, are they referring to an ensemble photo-oxidation study, or simply an inert multicomponent mixture? Did the authors consider a test intermediate in complexity, e.g. the obvious intermediate between a single compound mass spectrum and a chamber photo-oxidation experiment would be an analysis of a mixture of 2-3 compounds, without the complex oxidative chemistry. Was this considered?*

**Response**: This is a very good point and, yes, we did consider this. We are specifically referring to a range of mixed systems from inert multicomponent systems to those from additional chamber studies. The inert, or even reactive, multicomponent mixtures would enable us to better validate, and provide more training, to the tools presented here. This would give us increased, or decreased, confidence in the application to chamber systems. It would also enable us to perhaps construct a more generally applicable set of fingerprints to use in the training process.

*9) Regarding the AMS data employed (e.g. Figure 7): How were these data treated? Were they experiment averaged, summed, normalised? Despite the reference to Alfarra et al., 2013, it may be useful to briefly state this on introduction of the experimental data in order to provide context.*

**Response:** The data were normalised, as it was the relative peak contributions that were of interest; quantitative agreement on mass concentrations is a separate area of enquiry outside the scope of this work. We have also added a brief reference to the conditions mentioned in the Alfarra et al., 2013 study in section 3.3, page 9: *Figure 7 displays the predicted mass spectra for the GECKO-A model results of Valorso et al. (2011) combined with the experimental data taken from a chamber-based α-pinene SOA formation experiment reported by Alfarra et al. (2013). This spectra represents "aged" aerosol, after 4 hours of experiment, during which the VOC/NOx ratio was ~2.* The same information has been added to the caption of Figure 7.

*10) Please check reference formatting throughout, e.g. spaces between text and parentheses and improper use of chronological ordering of multiple citations.*

**Response:** Apologies, these formatting issues have been corrected.

*11) P2, Line 21: Please add more indicative primary source references; this paper is rather Specific*

**Response:** Apologies, we have now replaced this reference with the overarching review of Halquist et al. (2009).

12) P2, Line 30: Reference repeated

**Response:** Apologies, this has been corrected.

13) P3, Line 14: ": : :air and in THE laboratory: : :"

**Response:** Apologies, this has been corrected.

14) P4, Line 26: "now" rather than "new"

**Response:** Apologies, this has been corrected.

15) P4, Line 28: "than" rather than 'that"?

**Response:** Apologies, this has been corrected.

16) P5, Line 10: "than" rather than "that"

**Response:** Apologies, this has been corrected.

17) P5, Line 30: Full-stop missing after "3.2"

**Response:** Apologies, this has been corrected.

18) P6, Lines 14 – 16: Rewrite to facilitate ease of reading

**Response:** We have replaced those lines with the following:' *However we first and foremost wish to demonstrate the efficacy of using pre-defined fingerprints as they are available in the literature, or, within existing open-source software packages. The exact physical processes taking place within instrument are still the subject of considerable debate.* '

19) P8, Lines 9 – 11: Sentence is awkward, I suggest it is rewritten for clarity

**Response:** We have replaced those lines with the following:' *A recent study of McVay et al. (2016) presented results demonstrating sensitivity of aerosol mass and composition to processes included in a box-model model, including the addition of autoxidation mechanisms. They proposed that autoxidation might resolve some or all of measurement–model discrepancy from chamber simulations, but that this hypothesis could not be confirmed until more explicit mechanisms are established for α-pinene autoxidation(McVay et al., 2016).*'

20) P8, Line 21: "Fingerprints"

**Response:** Apologies, this has been corrected.

21) P9, Line 1: Repeated word - "value values"

**Response:** Apologies, this has been corrected.

22) P14, Tables 1 and 2 legends: Right [ parenthesis missing

**Response:** Apologies, this has been corrected.

23) P17, Figures 4: axis labels are too small and potentially unreadable in final print, please increase the text size

**Response:** Apologies, this has been corrected.

24) P17, Figures 4 legends: Right [ parenthesis missing

**Response:** Apologies, this has been corrected.

25) P19, Figure 5: Axis labels missing

**Response:** Apologies, this has been corrected.

26) P20, Figure 6: Axis labels missing

**Response:** Apologies, this has been corrected.